# Immune Actions on the Peripheral Nervous System in Pain

**DOI:** 10.3390/ijms22031448

**Published:** 2021-02-01

**Authors:** Jessica Aijia Liu, Jing Yu, Chi Wai Cheung

**Affiliations:** Laboratory and Clinical Research Institute for Pain, Department of Anaesthesiology, Queen Mary Hospital, The University of Hong Kong, Hong Kong 102, China; u3005063@connect.hku.hk

**Keywords:** peripheral nervous system, pain, immune response, inflammation

## Abstract

Pain can be induced by tissue injuries, diseases and infections. The interactions between the peripheral nervous system (PNS) and immune system are primary actions in pain sensitizations. In response to stimuli, nociceptors release various mediators from their terminals that potently activate and recruit immune cells, whereas infiltrated immune cells further promote sensitization of nociceptors and the transition from acute to chronic pain by producing cytokines, chemokines, lipid mediators and growth factors. Immune cells not only play roles in pain production but also contribute to PNS repair and pain resolution by secreting anti-inflammatory or analgesic effectors. Here, we discuss the distinct roles of four major types of immune cells (monocyte/macrophage, neutrophil, mast cell, and T cell) acting on the PNS during pain process. Integration of this current knowledge will enhance our understanding of cellular changes and molecular mechanisms underlying pain pathogenies, providing insights for developing new therapeutic strategies.

## 1. Introduction

Pain can be induced by tissue injury or different types of diseases that affect the somatosensory system, resulting in noxious (hyperalgesia) or non-noxious (allodynia) symptoms, which is an important defense mechanism to avoid harmful stimuli. Terminal nerves of somatosensory neurons (also known as nociceptors) innervate into the skin, cornea, internal organs, joints, bones, muscles, and deep visceral tissues, which are highly expressing a set of molecular sensors including transient receptor potential channel subtypes (TRPs), G protein coupled receptors (GPCRs) and sodium channel (Nav) [1,2]. Upon sensing noxious stimuli (e.g., mechanical, thermal and chemical), these nociceptors can quickly generate action potentials that are transmitted to the central nervous system (CNS) where the signals are processed. Nociceptor sensitization (or peripheral sensitization) at the site of the injury is therefore considered to be the primary cause of pain and the most appropriate targeting system for pain therapies [3,4].

Pain syndromes can be divided into acute and chronic stages. Acute pain plays a vital protective and adaptive role in warning the individual to avoid further injury and driving immune responses against infections or pathogens during healing. The inflammatory mediators produced by the immune system such as cytokines, lipid mediators, and growth factors directly activate nociceptive primary sensory neurons in the peripheral nervous system (PNS) evoking a pain response [4,5]. On the other hand, chronic pain is detrimental and arises from nerve damage caused by surgery, trauma, metabolic disorders (e.g., diabetic mellitus) or autoimmune diseases (e.g., multiple sclerosis) [6]. Chronic pain is a long-lasting syndrome and has substantial impacts on patients’ quality of life and high economic burdens on individuals and society [7]. Although alterations in the dorsal spinal cord and brain are one of the key mechanisms of chronic pain maintenance, peripheral sensitization is essential in the transition from the acute to the chronic stage [5,8]. Notably, emerging studies have revealed that bidirectional signaling between the immune and nervous systems contribute to the initiation and maintenance of chronic pain [2]. Altogether, our current knowledge of nociceptor–immune interactions have provided some molecular insights for developing better therapies for pathological inflammation-associated pain.

In this review, we discuss the interplay between immune cells (macrophages and/or monocytes, mast cells, neutrophils, and T cells) and the PNS in both acute and chronic pain, and their distinct functions in pain induction and modulation.

## 2. Peripheral Responses to Pain

Like the counterparts in the CNS, the PNS is also composed of neurons and glial cells, in which clusters of nociceptive sensory neurons are located in different ganglion in the trunk and the head that relay information about the environment to the CNS. The most common types of ganglion are the dorsal root ganglion (DRG) in the trunk, and others in the cranial including trigeminal and glossopharyngeal ganglia [9]. These sensory neurons (first-order primary afferent neurons) are classified to unmyelinated c-fibers and myelinated Aδ/β -fibers that transduce different types of pain signals, including mechanical, thermal, or chemical stimuli (Figure 1a). Free peripheral nerve endings function as receptive sites extend from neuronal cell bodies in the DRG or cranial nerve ganglion. Notably, their sensory neurons are pseudo-unipolar neurons that have one axon with two processes: one peripheral axonal branch innervates the tissues in the body to receive sensory information and the other axonal branch sends nerve impulses to excite second-order postsynaptic neurons in the dorsal horn of the spinal cord [3,5]. Subsequently, axons from second-order neurons project into thalamic nuclei in brain, where the third-order neurons transmit the pain signaling to the primary sensory cortex [10] (Figure 1b). The glial cells in the PNS mainly comprise Schwann cells and satellite glial cells (SGCs). The SGCs surround the somata of sensory neurons and usually consist of a single layer of cells connected to each other by gap junctions [11]. Schwann cells are the most abundant cell types in the PNS, which support axonal outgrowth by producing a variety of growth factors, such as nerve growth factor (NGF), glial cell line derived neurotrophic factor (GDNF), and brain-derived neurotrophic factor (BDNF) [12,13]. Schwann cells consist of two major phenotypes, myelinating Schwann cells and nonmyelinating Schwann cells [12]. Myelinating Schwann cells wrap larger axons in a 1:1 ratio to form the myelin sheath, nonmyelinating Schwann cells embed smaller axons, forming a remark bundle [13] (Figure 1c). 

After noxious stimuli, peripheral neurons/nerves and glial cells undergo significant pathological changes before central properties that contribute to the pain initiation and development through their interaction with immune signals. Noteworthy, signaling pathways between primary sensory neurons, SGCs, Schwann cells and immune cells are highly intertwined. For example, activated Schwann cells mediate the breakdown of the blood–nerve barrier via the secretion of matrix metalloproteinase 9 (MMP-9), which promotes the recruitment and infiltration of immune cells (e.g., macrophages and T cells) from the vasculature to the injury sites. [14,15]. Sensory neurons also produce neuropeptides at their peripheral endings that not only serve as attractions to induce the invasion of circulating immune cells but also modulate the activity of innate and adaptive immune cells [2,16]. A dense cluster of immune cells produce pronociceptive mediators directly acting on peripheral nociceptors to promote sensitization of pain signaling and the recruitment of immune cells and vice versa. In addition, reactive SGCs and immune cells (e.g., macrophages) work cooperatively to promote peripheral sensitization by releasing proinflammatory cytokines such as IL-1β, IL-6, and TNF [17,18,19,20,21]. In contrast, SGCs and macrophages are also involved in the regeneration of DRG axons and remyelination of Schwann cells [21,22]. The four major heterogeneous immune cell types (macrophages and/or monocytes, mast cells, neutrophils, and T cells) resident in or infiltrating the PNS have specific functions in pain modulation and sensitization, which are further discussed below.

## 3. Macrophages/Monocytes in the PNS

### 3.1. Recruitment and Activation of Monocytes and Macrophages

Monocytes are a population of plastic and heterogeneous blood cells that infiltrate the site of injury, where they alter their phenotype to either a proinflammatory or anti-inflammatory phenotype in response to environmental changes [23,24]. Upon injury or infection, monocytes can infiltrate into the DRG and sciatic nerve, where they differentiate into macrophages that exhibit increased staining for CD68. Both peripheral monocytes and macrophages play active roles in the modulation and pathogenesis of pain, with distinct mechanisms that are shaped by the causes and the context of pain [25,26,27,28,29]. In addition to monocyte-derived macrophages in nerve injury, there are also resident phagocytic macrophages that account for 9% of the cell populations in the PNS, which can rapidly migrate to the site of injury in response to stimuli [30,31]. The recruitment of macrophages is orchestrated by chemokine (C-C motif) ligands (CCLs) and inflammatory cytokines. The chemokine monocyte chemoattractant protein-1 (MCP-1), also known as CCL2, is significantly upregulated in small-diameter neurons of the DRG and induces peripheral sensitization via activation of its receptor CCR2 on TRPV1-expressing nociceptors [32,33]. In nerve injury or chemotherapy-induced neuroinflammation, the expression levels of CCL2 are increased, which promotes the infiltration and activation of CCR2-positive monocytes/macrophages in the DRG. Intrathecal administration of MCP-1 neutralizing antibodies reduced paclitaxel induced macrophage infiltration into the DRG [34,35,36]. Consistently, infiltration of macrophages to the nerve injury sites was significantly impaired in CCR2-deficient mice, indicating that CCL2 is a primary pro-recruitment molecule [37,38]. Chemokine (C-X3-C motif) ligand 1 (CX3CL1) was also reported to play a role in attracting macrophages. Levels of CX3CL1 were increased in the DRG after nerve injury, whereas blocking CX3CL1 inhibited the recruitment of macrophages to the DRG and attenuated allodynia in paclitaxel-induced neuropathy [39,40]. Moreover, CX3C chemokine receptor 1 (*CX3CR1*)-deficient mice exhibited significantly reduced monocytes/macrophages in both inflammatory and neuropathic pain models [29,38,40,41]. Notably, macrophage recruitment was not completely abolished even in the absence of chemokine signals, suggesting the involvement of other regulators such inflammatory cytokines. Shubayev et al. found the expression level of tumor necrosis factor alpha (TNF-α) was positively correlated with the number of macrophages in periphery nerves. Mechanistically, in response to stimuli, activated Schwann cells produce TNF-α that further induces matrix metalloproteases (MMP9) to disrupt the blood-nerve barrier facilitating the invasion from circulating immune cells including macrophages [14] (Figure 2). Consistently, *TNF-α* deficient mice with sciatic nerve transection exhibited poor macrophage recruitment [15]. Besides TNF-α, interleukin 15 (IL-15) can also activate MMP-9 expression. Intraneural administration of IL-15 into the sciatic nerve increased the infiltration of macrophages into the nerve and promoted the development of mechanical hyperalgesia [42,43,44]. Another proinflammatory cytokine, IL-1β, exhibited the highest levels around the site of injury even before the activation of macrophages, whereas administration of a function-blocking antibody against IL-1β into the injured sciatic nerve reverted the inhibited recruitment of macrophages and phagocytosis [45,46]. Interestingly, infiltrated macrophages can also release inflammatory mediators including CCL2, TNFα, IL-1α, and IL-1β, and in turn, these mediators can further boost up the attraction and recruitment of macrophages to the injury sites [46,47]. For example, Jia et al. found that paclitaxel could induce high level expressions of NLRP3 inflammasomes in the infiltrated macrophages within DRG and sciatic nerve, which promoted IL-1β production and mechanical allodynia in a chemotherapy-induced neuropathic pain model [48].

### 3.2. Pathological Role of Macrophages in Pain

The correlation between the infiltration of monocytes/macrophages and the development of pain is well established. However, the strongest evidence of the involvement of monocytes/macrophages in the induction or/and maintenance of pain comes from monocyte/macrophage depletion studies. Depletion of peripheral macrophages by intravenous injection of clodronate liposomes partially attenuated paclitaxel-induced or nerve injury-induced thermal and mechanical hyperalgesia in the DRG [26,36,49,50]. Additionally, monocyte/macrophage depletion with liposome-encapsulated clodronate (clophosome) delayed the progression of diabetes-induced mechanical allodynia [51]. Peng et al. ablated peripheral monocytes in transgenic mice in a temporally controlled fashion, which demonstrated the critical roles of infiltrated monocytes in initiating neuropathic pain and in promoting the transition from acute to chronic pain after peripheral nerve injury [52]. Notably, clophosome could not effectively ablate resident macrophages in the DRG, meaning the possible contribution of DRG macrophages cannot be ruled out in these studies. A recent study found that PNS-resident macrophages were self-maintaining and exhibited similar gene expression profiles to activated microglia in aging or neurodegenerative conditions [53]. Specifically, the depletion of macrophages in the DRG, but not at the peripheral nerve injury site, prevented the initiation of neuropathic pain and attenuated ongoing nerve-injury-induced mechanical hypersensitivity [54]. The cell-specific depletion of proliferating monocytes and macrophages in a model of inflammatory pain impaired the development of mechanical and thermal hypersensitivity caused by incision and pathogens, and also decreased proinflammatory mediators at the injury sites [28]. 

Production of proinflammatory mediators from monocytes/macrophages, such as cytokines (TNF-α, IL-1β), chemokines (CCL2), growth factors (NGF), and lipids (PGE2 and PGI2), is a key mechanism of nociceptor sensitization (Figure 2). Macrophage inflammatory protein (MIP)-1α, also known as CCL3, is an inflammatory chemokine secreted from macrophages, which promotes the development of neuropathic pain via the upregulation of IL-1β [55]. Mammalian toll-like receptors (TLRs), a family of 12 evolutionarily conserved membrane proteins expressed in macrophages, promote the synthesis of pro-inflammatory cytokines and chemokines upon activation [56]. The activation of TLR4 resulted in the release of both TNFα and IL-1β, whereas stimulation of TLR 3, 9, and 7 induced the production of IL-1α and 1β [56]. Interestingly, macrophage-derived TLR9 signaling showed sex-specific differences, with the promotion of chronic pain only in male mice, but not in female mice, in chemotherapy-induced peripheral neuropathy (CIPN) [57]. Neurotrophic growth factors, such as nerve growth factor (NGF), are key players in driving peripheral nerve sprouting and nociceptive signal transduction in both inflammatory and neuropathic pain. Macrophages are an important source of NGF production following inflammation and nerve injury [2,58]. In a model of arthritis, depletion of macrophage via clodronate liposomes reduced NGF production and pain behaviors [59]. On the other hand, macrophage-derived NGF was found to be substantially increased more than four weeks after injury, suggesting prolonged interaction between macrophages and nociceptive neurons might contribute to the maintenance of chronic pain [60]. Moreover, NGF not only acts on periphery nerve fibers, but also involves a feedback mechanism that increases membrane ruffling, calcium spiking, phagocytosis, and growth factor secretion by macrophages [61]. Stimulation of P2X4 receptors in macrophages triggered Cyclooxygenase (COX)-dependent release of lipid mediator prostaglandin E2 (PGE2), which regulates inflammation and pain hypersensitivity by promoting sensory neuron hyperexcitability [62]. Apart from regulating the release of pain mediators, macrophages are also involved in controlling metabolic factors. For instance, reactive oxygen species (ROS) could be generated by macrophages to engage TRPA1 in nociceptors [63]. Activation of macrophage angiotensin II type 2 receptor triggered neuropathic pain via oxidative stress and subsequently stimulated TRPA1 in nociceptors [64]. In the CIPN model, CX3CR1^+^ monocytes migrated to peripheral nerves, where they produced ROS to elicit pain via the activation of TRPA1 [29].

### 3.3. The Role of Macrophages in Modulating Pain 

Unlike the CNS, the neurons in the PNS have strong regenerative capacity via the recruitment and polarization of macrophages, a vital process for tissue repair [65]. Macrophages can be polarized into different phenotypes, including M1 and M2, resulting in distinct states of pain induction or modulation. The M1 macrophages exhibit a classically activated phenotype and produce proinflammatory mediators that promote pain, whereas the M2 macrophages are immunosuppressive cells that secrete anti-inflammatory cytokines and growth factors to promote tissue repair and resolution of pain [66,67]. Both M1 and M2 macrophages are maintained in a dynamic equilibrium and can undergo rapid phenotypic switching in response to signaling molecules in the microenvironment [68,69]. Nevertheless, proinflammatory macrophages have complex cytokine profiles and their role in the context of nerve injury is controversial. The M1 macrophages with pro-inflammatory features are often referred to as harmful, however, multiple studies suggest that these mediators play important roles in removing distal degenerating axons and myelin debris by phagocytosis, which enables the reorganization of Schwann cells and lays the foundation for the repair of injured axons [45,70,71,72,73]. In vitro, the exposure of Schwann cells and neurons to conditioned media from M1-primed macrophages appeared to enhance Schwann cell proliferation, reduce axonal outgrowth, and compromise neuronal survival [66,74].

In general, M1 macrophages undergo dynamic switching to M2 macrophages over time without increasing the number of subtypes. These M2-like macrophages can secrete anti-inflammatory cytokines such as IL-10, TGF-β, and specialized pro-resolving mediators (SPM) [75]. Moreover, depletion of monocytes prior to transient inflammatory pain induced by IL-1β or carrageenan prolonged the resolution of inflammatory pain from two days to over one week. The resolution of the inflammation-induced hyperalgesia was dependent on IL-10 produced by monocytes/macrophages [76]. A perineural injection of IL-4-induced M2 macrophages attenuated mechanical hypersensitivity via the release of opioid peptides including metenkephalin, dynorphin A, and b-endorphin [77], which further supported the existence of pain-modulating macrophages. Furthermore, activation of G protein-coupled receptor (GPR37) enhanced phagocytosis and promoted M2-like macrophage polarization with increased release of IL-10 and IL-1β, which helped to reverse the inflammatory pain. Consistently, macrophage depletion with clodronate delayed the recovery from heat hyperalgesia and mechanical allodynia in zymosan-induced inflammatory pain [78,79]. In a nerve injury model, macrophages were found to not only clear debris postinjury, but also played critical roles in regulating Schwann cell differentiation and remyelination of regenerated axons [80].

## 4. Neutrophils in the PNS

### 4.1. Recruitment and Activation of Neutrophils

Neutrophils are short-lived and mostly polymorphonuclear leukocytes that are generated by myeloid precursors in the bone marrow. After inflammation or damage, neutrophils are one of the first immune cells recruited to the affected tissues within a few hours, and are essential in the fight against infection, as well as clearing cellular debris in both septic and aseptic processes [81]. Early studies reported that leukotriene B4 (LTB4) and complement component 5a (C5a) recruited neutrophils to the site of inflammation to induce pain sensitization [82,83]. During neurogenic inflammation, noxious stimuli activate nociceptors that promote axon reflexes and generate action potentials that propagate through neighboring nerve terminals, triggering a rapid and local release of neuropeptides (SP, CGRP, VIP, or GRP) at peripheral branches. The neuropeptides released by sensory neurons serve as chemotactic effectors for neutrophil attraction, and also act as antimicrobicidal components [16,84,85]. Other mediators such as chemokines (CXCL1) and cytokines (TNF, IL-17) also function to facilitate chemotaxis to promote the infiltration and accumulation of neutrophils in peripheral nerves and the DRG [86,87,88] (Figure 2). On the contrary, a few studies revealed a suppressive role of nociceptors in recruiting and activating neutrophils in multiple models of infection and inflammation. In *S. pyogenes*-induced necrotizing fasciitis, ablation of nociceptors led to increased neutrophil recruitment together with more well-circumscribed abscesses, smaller necrotic lesions, and improved control of infection [89]. Similarly, cutaneous *S. aureus* infection activated nociceptor neurons resulting in mechanical and thermal hyperalgesia, whereas nociceptor ablation led to increased infiltration of neutrophils at the infection sites and in draining lymph nodes [90]. Baral et al. also revealed that nociceptor sensory neurons (TRPV1^+^) in the lung responded to noxious or harmful stimuli resulting in coughing, pain, and bronchoconstriction, whereas specific ablation of TRPV1^+^ neurons promoted the recruitment and surveillance of neutrophils, facilitating cytokine induction and lung bacterial clearance [91]. Consistently, *TRPV1^−/−^* mutants also exhibited increased neutrophil recruitment in a post-myocardial infarction inflammatory model [92]. The differential mechanisms underlying neutrophil recruitment and activity may be context dependent, where local immune responses related to nociceptors can lead to sterile inflammation or pathogenic inflammation. 

### 4.2. The Pathological Role of Neutrophils in Pain

Studies using various inflammatory pain models have attempted to reveal the function of neutrophils, although their roles are still controversial. Early reports suggested that neutrophils can promote the sensitization of primary afferent neurons by releasing cytokines and chemokines that act as a positive feedback to activate neutrophils, triggering a complementary alternative pathway to amplify nociceptive responses [10,83]. Inhibition of neutrophil recruitment by fucoidin attenuated mechanical allodynia in carrageenan-induced inflammatory pain. Depletion of neutrophils with vinblastine sulfate or anti-neutrophil antibody decreased mechanical hyperalgesia in incision models [87]. Similarly, C5a-induced hypernociception was reduced in neutrophil-depleted rats [83]. In humans, neutrophils increasingly accumulate in the joints of patients with inflammatory pain such as arthritis, and their recruitment was found to be associated with the development of hyperalgesia. However, antibody-induced neutropenia had no effects on the mechanical and thermal hypersensitivity in complete Freund’s adjuvant (CFA)- and zymosan-induced pain [93,94]. By depleting different populations, Ghasemlou et al. revealed that lymphocyte antigen 6 complex locus G (Ly)6G^+^CD11b^+^ neutrophils did not contribute to the development of thermal or mechanical pain hypersensitivity in either incisional pain or a CFA-induced inflammatory model, whereas proliferating CD11b^+^Ly6G^−^ myeloid cells were necessary for mechanical pain hypersensitivity [28]. 

In neuropathic pain models, robust infiltration of neutrophils was observed at peripheral nerve injury sites peaking within the first few hours after damage, whereas neutrophils were barely detectable in intact nerves. Neutrophils release mediators that sensitize nociceptors in acute stage and recruit immune cells (e.g., macrophages and T cells) to the injury site, where they secrete a myriad of proinflammatory mediators that maintain the neuropathic pain [95,96]. Depletion of circulating neutrophils reduced the development of thermal hyperalgesia in a partial sciatic nerve transection model [97]. Genetic ablation of mediators or receptors regulating neutrophil adhesion and migration improved mechanical hyperalgesia in experimental neuropathic pain [98,99]. In a chronic constriction injury model, damaged peripheral nerves induced the migration of neutrophils to the ipsilateral side of the DRG, resulting in the release of chemokine CCL2 leading to peripheral nociceptor sensitization [100]. Depletion of circulating neutrophils by an anti-neutrophil antibody during injury in a herpetic neuralgia model significantly attenuated hypersensitivity via the inhibition of TNF in the DRG [101] (Figure 2). 

### 4.3. Neutrophils in the Resolution of Pain 

Neutrophils at the site of inflammation can secrete analgesic mediators such as opioid peptides (b-endorphin, met-enkephalin, and dynorphin-A), which can activate opioid receptors on peripheral sensory neurons resulting in the inhibition of nociceptive transmission [102]. Local injection of corticotropin releasing factor (CRF) evoked the release of opioid peptides from neutrophils to attenuate CFA-induced inflammatory-hyperalgesia in rat [103]. Interestingly, Lindbord et al. found that neutrophils can function similarly to infiltrating monocyte-derived macrophages during Wallerian degeneration (WD), which is an essential preparatory process for the axonal regeneration. Ablation of neutrophils substantially attenuated myelin removal after peripheral injury in both WT mice and CCR2 null mutants, replacing CCR2+ microphages as the primary phagocytic cells [104]. In addition, neutrophils can induce macrophage polarization by suppressing NF- κB activation upon injury, exerting anti-inflammatory effects [105]. 

## 5. Mast Cells in the PNS

### 5.1. Location of Mast Cells in PNS

Mast cells are granulated immune cells that reside in close proximity to nerve fibers in connective tissues, particularly CGRP^–^ and substance P^+^ neurons [106,107]. Within 24  h following injury, mast cells participate in innate host defense and immune reactions via degranulation to release a broad range of proinflammatory cytokines and chemokines [108]. Due to the unique localization of mast cells, it has long been postulated that a tight interaction between peripheral nociceptors and mast cells is involved in pain induction under different contexts. It has been shown that N-cadherin plays an essential role in the synapse-like structures involved in peripheral nerve terminal-mast cell communication [109]. Folgueras et al. found that metalloproteinase MT5-MMP (MMP-24) promoted the interaction between sensory fibers and mast cells via regulating N-cadherin, which led to thermal nociception and inflammatory hyperalgesia [110]. In humans, increased numbers of nerve mast cells were frequently reported in inflammatory diseases, and could be correlated to the severity of pain symptoms in patients with arthritis, allergic disease, and fibromyalgia [111,112,113]. 

### 5.2. The Role of Mast Cells in Pain

The strongest evidence of the involvement of mast cells in regulating peripheral sensitizations comes from depletion studies. Mast cell-deficient mice (*Kit*W-sh/W-sh) were hyporesponsive to vertically applied punctate heat stimuli, and had attenuated pelvic pain associated with neurogenic cystitis [114,115]. In rodents, administration of a secretagogue compound 48/80 induced degranulation of mast cells and caused immediate hyperalgesia and excitation of meningeal nociceptors, whereas mast cell-deficient mice (*Kit*W-sh/W-sh) could abrogate thermal and mechanical hyperalgesia induced by secretagogue compound 48/80 [116]. Mast cells have also been found to be important contributors to the development of cutaneous and deep hyperalgesia, and hypoxia-reperfusion-induced pain in transgenic sickle mice primarily through promoting the release of neuropeptides [117]. Mechanistically, degranulation of mast cells triggers the rapid secretion of histamine, serotonin, nerve growth factor, cytokines, and leukotrienes upon exposure to stimuli, which exert their various effects on nervous system or/and immune cells contributing to pain sensitization [118,119,120,121]. For example, histamine can promote release of neuropeptides (e.g., CGRP) and glutamate (excitatory neuronal transmitter) from nerve endings. Administration of antagonists of histamine receptors (H1–H5) reduced mechanical and thermal hyperalgesia as well as inflammation responses [122]. Particularly, antagonists targeting H2 receptor have been proved to be an effective treatment for painful bladder syndrome under clinical trials [123]. On the contrary of the analgesic effects of serotonin (5-HT) in CNS, 5-HT produced from mast cells in periphery contributes to the generation and maintenance of pain [124,125]. Growth factor, NGF, synthesized by mast cells, can directly bind to its receptor TrKA in TRPV1 neurons, evoking thermal and mechanical allodynia. In turn, NGF stimulates mast cells to release pronociceptive mediators, which results in a noxious microenvironment facilitating chronic hypersensitivity [126,127]. 

### 5.3. Recruitment and Degranulation of Mast Cells

The degranulation of mast cells and release of inflammatory mediators can be evoked by the activation of a variety of cell surface receptors, such as Fcɛ receptors and G-protein -coupled receptors [128]. It has been shown that Fcɛ receptors activate sphingosine kinases to promote generation and secretion of sphingosine-1-phosphate (S1P), a ligand for G-protein-coupled receptors, from mast cells. S1P plays critical roles in recruiting mast cells to the injury sites and further enhancing their degranulation in an autocrine manner by directly binding to G-protein-coupled receptors [129]. Notably, a few antagonists targeting S1P receptors have been developed, exhibiting strong analgesic effects in multiple neuropathic pain models [130,131]. Other studies found the MAS-related G protein-coupled receptor member B2 (*Mrgprb2*) was highly enriched in mast cells, and *Mrgprb2* mutant mice exhibited abolished mast cell activation by secretagogues [132,133]. Using a postoperative pain and CFA-induced pain model, Green et al. demonstrated that the mast cell-specific receptor Mrgprb2 mediated mechanical and thermal hyperalgesia and was required for the recruitment of innate immune cells at the injury site. Mechanistically, neuropeptide substance P (SP), an endogenous agonist of Mrgprb2, was able to activate human mast cells, leading to the release of multiple proinflammatory cytokines and chemokines via the human homolog MRGPRX2, which facilitates immune cell infiltration. Surprisingly, the SP-mediated inflammatory responses were independent of its canonical receptor, neurokinin-1 receptor (NK-1R). These results identified Mrgprb2/X2 as an important neuroimmune modulator and a potential target for the treatment of inflammatory pain [134] (Figure 2). In the enteric nervous system, CGRP released from enteric sensory neurons promoted the activation and degranulation of mast cells leading to intestinal immune diseases [135]. Similarly, CGRP-mediated vasodilation and mast cell degranulation were found to be key mechanisms in driving migraine in a mouse model [136]. Pannexin 1 (Panx1) is a large-pore membrane channel expressed in neurons, glial cells and immune cells, which promotes central or primary sensitizations upon the activation [137,138]. A recent study revealed that Panx1 plays a critical role in the degranulation of mast cells during hypersensitivity reaction promoted by ovalbumin, whereas the absence of Panx1 prevented histamine release or sustained Ca^2+^ signal increase [139]. It is well noted that pharmacological inhibition or genetic ablation of *Panx1* attenuated hypersensitivity in nerve injury models [138,140]. However, whether Panx1 contributes to the degranulation of mast cells in peripheral nerves remains to be determined.

## 6. T cells in the PNS

### 6.1. Subtypes of T Cells

The T cell is a type of lymphocyte originating from hematopoietic stem cells, and is characterized by the expression of a unique surface molecule, T cell receptor (TCR). Similar to other immune cells, T cells infiltrate into the sciatic nerve and dorsal root ganglia (DRG) after peripheral nerve injury, where they induce the release of proalgesic mediator leukocyte elastase (LE), leading to mechanical allodynia [25,120,141,142,143]. T cells can be grouped into different subtypes including T-helper cells (CD4^+^) and cytotoxic T cells (CD8^+^), which exhibit distinct functions in regulating pain. T-helper cells contain type 1 (Th1), type 2 (Th2), type 17 (Th17), and regulatory T cells (Treg cells), which play roles in modulating the innate and adaptive immune response. Each subsets of T cells are exhibiting unique transcriptional factors and cytokine production profiles. For example, Th1 cells express STAT4 and release IL-2/INFγ; Th2 cells express GATA3 and promote release of IL-4 and IL-10; Th17 cells express RoRγT and release IL-17; and Treg express FOXP3 and produce TGF β and IL-10 [144]. 

### 6.2. Recruitment and Polarization of T Cells

Neurotransmitters (glutamate) and neuropeptides (CGRP, SP, VIP) released by nociceptors are key mediators of T cells. These mediators are expressed at peripheral nerve endings in response to stimuli, and significantly affect the adaptive immune response including T cells. Silencing of nociceptors interrupted neuro-T cell interactions and inhibited amplified adaptive immune responses [90,145,146]. Similar to neutrophils, infiltrating T cells come from the endoneurial vasculature and rely on recruited phagocytic cells (e.g., marcophages), as ablation of these cells by clodronate lipsome prevented T cells (CD4+) invading the injury sites. These studies suggested that early stage recruitment and activation of other immune cells are prerequisite for T cell infiltration [27,28]. It is interesting to note that activation of distinct receptors expressed in T cells (e.g., inotropic and metabotropic glutamate, SP, and CGRP receptors) can drive infiltration and the differential phenotypic polarization of T cells in response to pain and immune defense against infection [147]. In an in vitro study, coculture of naive CD4^+^ T cells with superior cervical ganglion neurons favored Foxp3^+^ Treg cell polarization, which produce immunoregulatory cytokines, TGF- β, and IL-10. The study also reported that the generation of Tregs could be induced by neuropeptide CGRP released from nociceptive neurons [147]. Activation of CGRP inhibited Th1 cells, but promoted Th2 cells in contact hypersensitivity of the skin [148]. In a mouse model of inflammatory arthritis, SP was also shown to drive the polarization of Th17 cells, a subset of effector memory T cells that produce IL-17 [149,150]. 

The interaction between T cells and the nervous system is bidirectional. For example, vasoactive intestinal peptide (VIP) drives CD4^+^ T cells to produce proinflammatory cytokines including IL-5, and in turn, IL-5 activates sensory neurons to enhance sensitizations [146]. Meningeal T cells can trigger the production of brain-derived neurotrophic factor (BDNF) via the secretion of IL-4, which promotes brain neurogenesis [151]. The secretion of IL-31 by Th2 cells could directly activate receptors on sensory neurons inducing itch neuronal hyperexcitation in an inflammatory skin disease model [152]. 

### 6.3. The Role of T Cells in Pain

Many studies have revealed the distinct roles of T cell subtypes in regulating neuropathic pain. Early studies reported that Th1 cells promoted pain hypersensitivity by releasing proinflammatory cytokines (IL-1β, TNF-α, and IL-17), whereas type 2 cells reduced mechanical allodynia and thermal hyperalgesia by producing anti-inflammatory cytokines (IL-4 and IL-10) in neuropathic models [95,144]. Consistently, reduction of T cells markedly attenuated hyperalgesia and allodynia induced by nerve injury in rodent models [95,153]. Adoptive transfer of toxic T cells via intrathecal injection enhanced pain sensitization, whereas injection of Treg cells attenuated neuropathic pain following chemotherapy [154]. It is interesting to note that naïve DRG contains low populations of both CD4+ and CD8+ T cells, with a greater proportion of the latter [141,154,155]. The T cells infiltrating to the DRG after injury are mostly CD4+, suggesting a shift to CD4+ from a predominantly CD8+ population [142,156]. Consistently, genetic ablation of CD4+ T cells in mice or intravenous administration of CD4 antibodies reduced hyperalgesia and allodynia following neuropathic pain induction, which could be abolished by adoptive transfer of CD4+ T cells [27,157]. 

In inflammatory pain models, T cells appeared to be dispensable or even beneficial to the pain response. Intraplantar CFA injections significantly increased T cells in the inflamed tissue, whereas T cell-deficient mice (*Tcrb*^−/−^, *Tcrd*^−/−^, *Rag1*^−/−^, and *Rag2*^−/−^) showed no change in pain hypersensitivity, suggesting that T cells are not involved in the development of inflammatory pain [28,158,159]. Consistently, no significant differences in thermal or mechanical allodynia was found between *Tcrb*^−/−^ or *Tcrd*^−/−^ mutants and WT mice in a postoperative pain model [28,159]. Moreover, T cells facilitated the resolution of pain response after inflammation. A few studies revealed the deficiency of T cells resulted in prolonged mechanical allodynia in mice [160,161,162,163]. In addition, T cell-deficient mice (*Cd-1* nude, *Rag1* null mutant, and *Cd-4* null mutant) had pronounced deficiencies in opioid-mediated analgesia with increased hypersensitivity, whereas restoration of CD4^+^ T cells in *Rag1^−/−^, Rag2^−/−^,* or nude mice normalized the resolution of inflammatory pain [164]. In antigen- and collagen-induced models of arthritis, the depletion of CD8^+^ T cells worsened pain hypersensitivity [165]. 

## 7. Clinical Implications and Future Perspectives

Pain syndromes frequently occur in various disease conditions causing negative impacts on patients’ quality of life both physically and physiologically. Mounting evidence shows that neuroimmune interactions play critical roles in peripheral sensitization, which is highly associated with the initiation and maintenance of pain, and the transition from acute to chronic pain. Modulation of these interactions and mediators have been considered as promising strategies for the treatment of pain and the underlying diseases. Despite preclinical and clinical studies of a number of immune mediators that act on nociceptor sensitization and pain responses, little knowledge has been translated into pain-relieving therapies. For example, CCR2 or CSF1R antagonists have been applied to target myeloid cells for treating neuropathic pain, but little improvement was observed [166]. Moreover, TRPV1 antagonists exhibited adverse effects such as hyperthermia, and NGF antagonists attenuated pain in animal models, but caused nerve damage and joint destruction [167]. Targeting macrophages can be even more challenging, as there are at least two distinct subpopulations, M1 or M2, which play critical roles in both promoting and resolving pain, respectively. Similarly, N1 or N2 neutrophils have potential roles in stimulating and suppressing immunity. Currently, there are no specific molecular markers that have been isolated in these subpopulations for study. It is important to point out that acute inflammation not only induces pain, but can also promote the resolution of pain by triggering immune responses against pathogens and by producing specialized pro-resolving mediators (SPMs). 

In future studies, the identification of specific targets of neuroimmune interactions and their mediators that act on nociceptors could offer alternative pain treatments without off-target or adverse effects. Neuroimmune signaling is a double-edged sword under different contexts, which presents several big challenges: (1) How do we modulate immune signaling molecules without altering neuroprotective and neuroregenerative potential? (2) Can we remodel pathological immune actions to be neuroprotective and neuroregenerative by nuance manipulation? A greater and more precise understanding of mechanisms underlying neuroimmune signaling in humans is warranted, especially in clinical populations suffering from chronic pain.

## Figures and Tables

**Figure 1 ijms-22-01448-f001:**
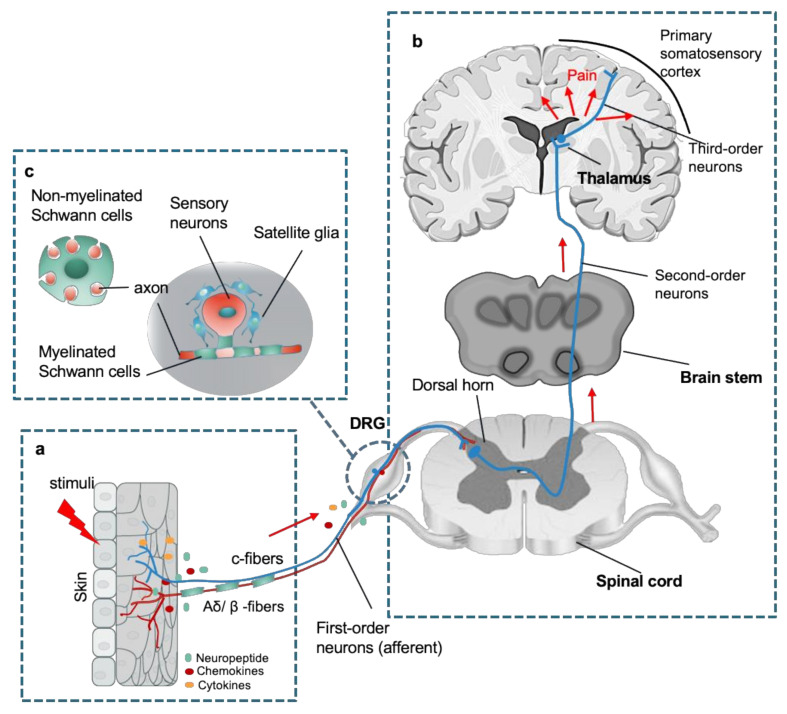
An overview of peripheral nervous system (PNS) in the sensory pathways leading from the skin to the brain. (**a**) The peripheral nerve endings comprise both unmyelinated c-fibers and myelinated Aδ/β -fibers in the skin that sense a stimulus, where chemicals such as inflammatory mediators and neuropeptides released from the injury site or the nerve endings activate the receptors and channels on the adjacent peripheral nerve terminals, subsequently resulting in initiating an action potential at the initial segment of the axon. (**b**) The axon of the peripheral sensory neuron (first order neuron) enters the spinal cord and contacts second-order neuron in the gray matter, where an action potential is generated at the initial segment of this neuron and travels up the sensory pathway to a region of the brainstem and thalamic nuclei. The sensory signal reaches the third-order neurons from the thalamus, and these project pain signaling to several cortical and subcortical regions (red arrows). (**c**) Schematic of organization of dorsal root ganglion (DRG). Sensory neuronal bodies are separated and wrapped satellite glial sheath. Myelinated Schwann cells envelop large diameter axons of sensory neurons, whereas nonmyelinated Schwann cells ensheath small diameter axons forming a remark bundle (upper left in (**b**)).

**Figure 2 ijms-22-01448-f002:**
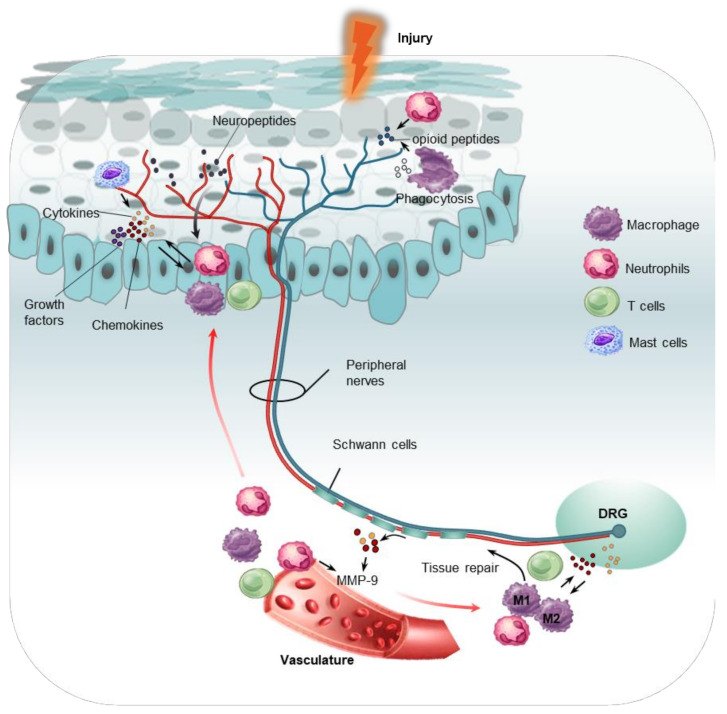
Interactions between distinct parts of the PNS with immune cells (neutrophils, macrophages, mast cells and T cells). Upon stimuli, injury initiates the release of inflammatory mediators (cytokines and chemokines) that cause degranulation of mast cells close to the nerve terminal, resulting in proinflammatory production. Nociceptive nerve terminals secrete neuropeptides through antidromic activation of neighboring nerve terminal branches to attract immune cells. Activated Schwann cells and neutrophils mediate breakdown of the blood–nerve barrier via the secretion of matrix metalloproteinase 9(MMP-9), promoting infiltration of immune cells including macrophages, T cells to the DRG and peripheral nerve ending. These cells, once activated, release a battery of inflammatory mediators (growth factors, cytokines, and chemokines) that act on receptors expressed on adjacent nociceptor nerve terminals, leading to nociceptor sensitization. Macrophages exhibit functions in mediating phagocytosis and tissue repair. Both macrophages and neutrophils inhibit nociceptive effects by releasing opioid peptides at injury sites.

## Data Availability

Not applicable.

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
