# Peer review of "Immune Actions on the Peripheral Nervous System in Pain"

_ijms, 2021, doi:10.3390/ijms22031448_

Round 1
Reviewer 1 Report
The review is well designed and performed. Overall, the study will be of high interest to the field of clinical neuroscience and open new perspectives in the development of new drugs targeting to treatment of chronic pain. However, I want the authors address the following comments.
Minor comments:
Comment 1: Line 29 and 30 of the introduction repeat the word "muscle".
Comment 2: The authors comment the importance of the involvement of IL-1β in recruitment and activation of monocytes and macrophages. Could be interesting that the authors to comment on the involvement of the NLRP3 inflammasome in macrophage [1].
Comment 3: The authors describe the recruitment and degranulation of mast cells in pain. Has been described a possible mechanisms of the participation of P2X7R and pannexin 1 in chronic pain in the central nervous system [2]. A key role for the pannexin 1 in neuropathic pain has been advocated because it is widely-expressed in neurons, glia and mast cells and because pannexin 1 can be activated by various mechanisms that are relevant to nerve injury [3]. It will be nice that the author can address this matter in manuscript.
[1] Jia, M., Wu, C., Gao, F., Xiang, H., Sun, N., Peng, P., ... & Tian, B. (2017). Activation of NLRP3 inflammasome in peripheral nerve contributes to paclitaxel-induced neuropathic pain. Molecular pain, 13, 1744806917719804.
[2] Bravo, D., Maturana, C. J., Pelissier, T., Hernandez, A., & Constandil, L. (2015). Interactions of pannexin 1 with NMDA and P2X7 receptors in central nervous system pathologies: Possible role on chronic pain. Pharmacological research, 101, 86-93.
[3] Weaver, J. L., Arandjelovic, S., Brown, G., Mendu, S. K., Schappe, M. S., Buckley, M. W., ... & Krupa, J. (2017). Hematopoietic pannexin 1 function is critical for neuropathic pain. Scientific reports, 7, 42550.
Author Response
Reviewer #1
Comment 1: Line 29 and 30 of the introduction repeat the word "muscle".
Response: We have deleted the duplicated word “muscle” in line 29.
Comment 2: The authors comment the importance of the involvement of IL-1β in recruitment and activation of monocytes and macrophages. Could be interesting that the authors to comment on the involvement of the NLRP3 inflammasome in macrophage [1].
Response: We have discussed the involvement of NLRP3 inflammasome in macrophages and neuropathic pain. This part was described in line 155-157.
Comment 3: The authors describe the recruitment and degranulation of mast cells in pain. Has been described a possible mechanisms of the participation of P2X7R and pannexin 1 in chronic pain in the central nervous system [2]. A key role for the pannexin 1 in neuropathic pain has been advocated because it is widely-expressed in neurons, glia and mast cells and because pannexin 1 can be activated by various mechanisms that are relevant to nerve injury [3]. It will be nice that the author can address this matter in manuscript.
Response: We have discussed the role of pannexin channel in promoting hypersensitivity and mast cell degranulation. This part was described in line 371-378.

Reviewer 2 Report
This review provides a good overview of the relationship between peripheral nervous system and immune cells in pain disorders. It also covers important research papers and appropriately presents current challenges for therapeutic applications.
However, there are a fair number of grammatical or spelling mistakes that should be corrected, especially Remark bundle, which is a Remak bundle mistake.
Author Response
Reviewer #2
Comment 1: However, there are a fair number of grammatical or spelling mistakes that should be corrected, especially Remark bundle, which is a Remak bundle mistake.
Response: We have carefully checked grammatical or spelling errors and revised the mistakes accordingly. Track changes were kept for visualization.
